# Size-dependent vitrification in metallic glasses

Valerio Di Lisio [1], Isabella Gallino [2] ✉, Sascha Sebastian Riegler[2], Maximilian Frey [2], Nico Neuber [2], Golden Kumar [3], Jan Schroers [4], Ralf Busch [2] & Daniele Cangialosi [1,5] ✉

Reducing the sample size can profoundly impact properties of bulk metallic glasses. Here, we systematically reduce the length scale of Au and Pt-based metallic glasses and study their vitrification behavior and atomic mobility. For this purpose, we exploit fast scanning calorimetry (FSC) allowing to study glassy dynamics in an exceptionally wide range of cooling rates and frequencies. We show that the main $\alpha$ relaxation process remains size independent and bulk-like. In contrast, we observe pronounced size dependent vitrification kinetics in micrometer-sized glasses, which is more evident for the smallest samples and at low cooling rates, resulting in more than 40 K decrease in fictive temperature, $T_f$, with respect to the bulk. We discuss the deep implications on how this outcome can be used to convey glasses to low energy states.

The transformation of a liquid, supercooled below its melting temperature, into a glass, the so-called vitrification or glass transition, remains one of the most intriguing unsolved problems in condensed matter physics[1,2]. Apart from the underlying fundamental understanding, the way vitrification takes place and the age of the glass[3–5] can deeply impact the glass properties and their lifetime evolution[6,7]. Among the variety of glasses, a class of utmost importance from both fundamental and technological viewpoints is that of metallic glasses (MG). They combine different technological relevant properties such as superior mechanical properties and corrosion resistance[8,9], which could be deeply affected by how vitrification has previously taken place[10]. For instance, the fracture toughness of MG has been directly related to the enthalpic state of the glass[11,12], which can be varied by vitrifying at different cooling rates and described by the concept of fictive temperature[13], $T_f$. The latter is defined as the temperature at which the glass line drawn from the glass thermodynamic state crosses the supercooled liquid line or, for experiments conducted on cooling, simply the temperature at which the supercooled liquid falls out of thermodynamic metastable equilibrium[14].

Conventional wisdom describes vitrification as triggered exclusively by the main, $\alpha$, relaxation exhibiting super-Arrhenius temperature dependence[15–20]. In this case, the cooling rate-dependent $T_f$ exhibits the same behavior as the temperature-dependent relaxation time, $\tau$, of the $\alpha$ relaxation. Recently, it has been shown that, for an Au-based MG, the cooling rate-dependent vitrification takes place with a weaker temperature dependence than the $\alpha$ relaxation[10]. As a consequence, vitrification at low cooling rates results in values of $T_f$ lower than those which would be obtained only accounting for the $\alpha$ relaxation. The main consequence of this outcome is that atomic mechanisms different from the $\alpha$ relaxation may be important actors in vitrification kinetics[21,22].

Beside the role of different mechanisms in vitrification kinetics, a long-standing problem concerns whether vitrification and atomic/molecular relaxation can be modified by reducing the sample size. Early work on low molecular weight glass formers o-terphenyl (OTP) and benzyl alcohol confined in nanopores showed a depression of the glass transition temperature, $T_g$, with respect to the bulk at nanopores sizes below ~70 nm[23]. Later, Keddie et al.[24] pioneered studies on the effect of thickness on the $T_g$ of thin polymer films, showing a reduction

[1]Donostia International Physics Center, Paseo Manuel de Lardizabal 4, 20018 San Sebastián, Spain. [2]Saarland University, Chair of Metallic Materials, Campus C6.3, 66123 Saarbrücken, Germany. [3]Department of Mechanical Engineering, University of Texas at Dallas, Richardson, TX, USA. [4]Yale University, Mechanical Engineering and Materials Science, New Haven, CT, USA. [5]Centro de Física de Materiales (CSIC–UPV/EHU), Paseo Manuel de Lardizabal 5, 20018 San Sebastián, Spain. ✉e-mail: i.gallino@mx.uni-saarland.de; daniele.cangialosi@ehu.eus

of $T_g$ for an archetypal polymeric glass former, polystyrene (PS), supported on inorganic substrates for thicknesses smaller than about 50 nm. The intense activity, conducted during the last decades and summarized in several reviews[25–28], depicted a scenario where effects on the $\alpha$ relaxation, if any, are present for samples with typical size below 10 nm[29–31]. Consistently, electron correlation microscopy with sub-nanometer resolution showed that the dynamics of MGs at the free surface are perturbed at length scales not exceeding a few nanometers[32,33]. In stark contrast, effects on vitrification kinetics are present for much larger sample sizes and, for polymeric glasses not exposed to adsorbing interfaces[27], generally result in significant $T_g$ reduction with respect to the bulk. When the concept of $T_f$ is employed to characterize vitrification kinetics, significant reductions have been found for polymer samples with size exceeding the micrometer length scale[34,35]. While, in view of the invariant bulk-like $\alpha$ relaxation any explanation invoking any change of dynamics at such large length scale must be discarded, significant effort has been undertaken to explain this finding on the base of a model of equilibration based on diffusion of free volume holes[26]. A direct visualization of the physical soundness of this model was provided in colloidal glasses[36], where the migration and annihilation of free volume holes at the interface were visualized by microscopy.

While deeply investigated in glass-forming polymers, whether reducing the sample size in MG may be of relevance in affecting vitrification kinetics has so far remained completely elusive and largely unexplored. This might have profound implications for the macroscopic properties of the MG. For example, the ordinary temperature previously associated with a ductile-to-brittle transition in MG was recently proved to play a secondary role[37], and the concept of $T_f$ dependent mechanical properties of MG is emerging[11,12,37–39]. One of the most intriguing observations is that a gradual change in fracture morphology of a Pt-based MG from vein-pattern to completely smooth fracture surface to necking is observed with decreasing sample size at micrometer length scales and testing temperature[39].

In this work, we investigate size-dependent glass transition in two archetypal MG, based on Au and Pt, respectively. For this purpose, we employ fast scanning calorimetry (FSC) permitting a large range of heating/cooling rates from 0.5 K s$^{-1}$ up to 5000 K s$^{-1}$. In samples prepared in identical conditions, we investigate both atomic mobility[40,41], that is the rate of spontaneous fluctuations taking place in the unperturbed supercooled liquid at equilibrium, and vitrification kinetics[20] in a wide range of time scales. We find that atomic mobility remains bulk-like for all investigated sample sizes, ranging from bulk to several microns. In contrast, we observe pronounced size-dependent vitrification kinetics more evident for the smallest samples and at low cooling rates. As a result, vitrification of MG takes place at temperatures lower than bulk for samples size below ~10 μm. The important implication of this outcome is that mild reductions of the sample size in MG allow exploring thermodynamic states, in terms of $T_f$, deep down in the energy landscape, thus opening the door to the obtainment of thermodynamically ultra-stable MG in time scales amenable to the experimental practice.

## Results

We begin presenting results obtained by using step-response protocols. By applying a linear perturbation, that is, a small temperature change, this methodology conveys information on the time scale of spontaneous fluctuations via thermal susceptibility. Once Fourier transformation from the time to the frequency domain is carried out, step-response protocols deliver the complex specific heat: $c_p^* = c_p' + ic_p''$. Frequency-dependent reversing specific heat, $c_{p,rev}$, that is, the modulus of $c_p^*$, which approximately equals the in-phase specific heat, $c_p'$, is presented in Fig. 1. The main $\alpha$ relaxation is identified as a step in $c_{p,rev}$. The insets of Fig. 1 show frequency and temperature dependent $c_{p,rev}$ for both glasses. As customary, increasing the frequency results in a

temperature up-shift of the step in $c_{p,rev}$, implying an acceleration of the $\alpha$ relaxation with increasing temperature. Furthermore, a broad tail in excess with respect to the glassy specific heat, more visible at low frequencies and for Au$_{49}$Cu$_{26.9}$Si$_{16.3}$Ag$_{5.5}$Pd$_{2.3}$ at. %, is detected, which can be ascribed to a secondary relaxation[10,42].

The main panels of Fig. 1 underline the effect of sample size showing temperature-dependent $c_{p,rev}$, obtained exemplary at 20 Hz, for both alloy compositions with different $l_{eq} = V/A$, where $l_{eq}$ is the equivalent size, and $V$ and $A$ are the sample volume and surface area, respectively; obtained analyzing samples SEM micrographs. Here, as detailed in the Methods section, $V$ and $A$ were obtained from the diameter in spherical samples. In the case of film-like samples, while $A$ was visually determined from SEM micrographs, $V$ was obtained from the knowledge of the sample mass via the alloy density. Irrespective of the sample size, the step in $c_{p,rev}$ takes place in the same temperature range, indicating the absence of a size dependence on the MG atomic mobility. The mid-step of $c_{p,rev}$ defines a typical relaxation time, $\tau$, of the $\alpha$ relaxation, whose temperature dependence is presented in Fig. 3a, b. The size independence of the $\alpha$ relaxation observed at 20 Hz

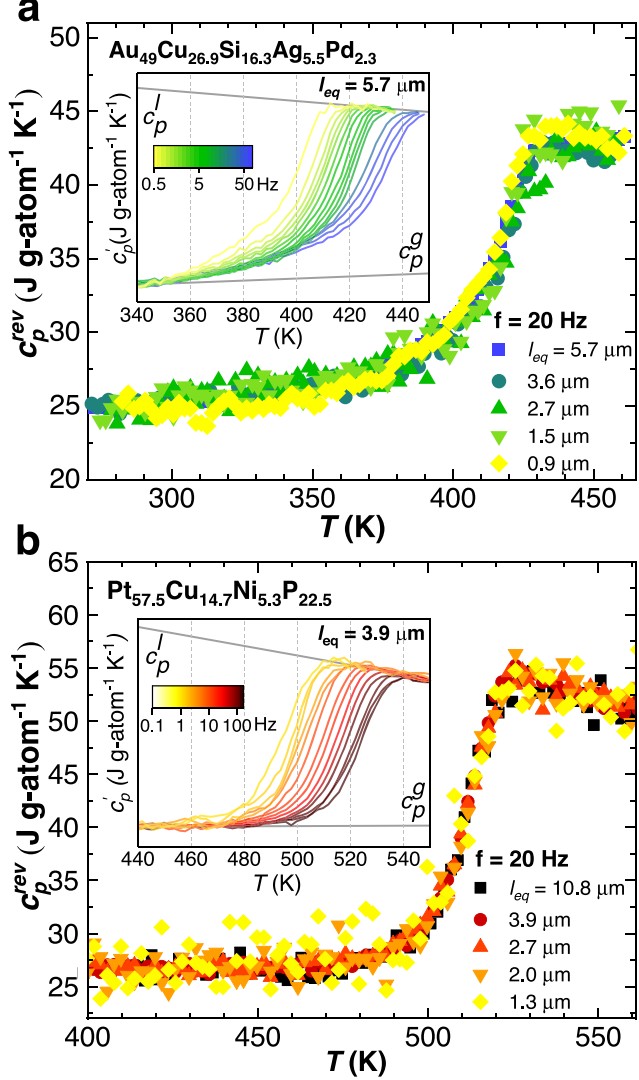

**Fig. 1 | Linear response in terms of thermal susceptibility for metallic glasses of different size.** Reversing specific heat at 20 Hz for samples with different characteristic lengths, $l_{eq}$ (main panels), and for samples with the indicated $l_{eq}$ at different frequencies (insets) as a function of temperature for **a** Au$_{49}$Cu$_{26.9}$Si$_{16.3}$Ag$_{5.5}$Pd$_{2.3}$ at.% and **b** Pt$_{57.5}$Cu$_{14.7}$Ni$_{5.3}$P$_{22.5}$ at.%. The gray lines are linear fits to glass and liquid-specific heats.

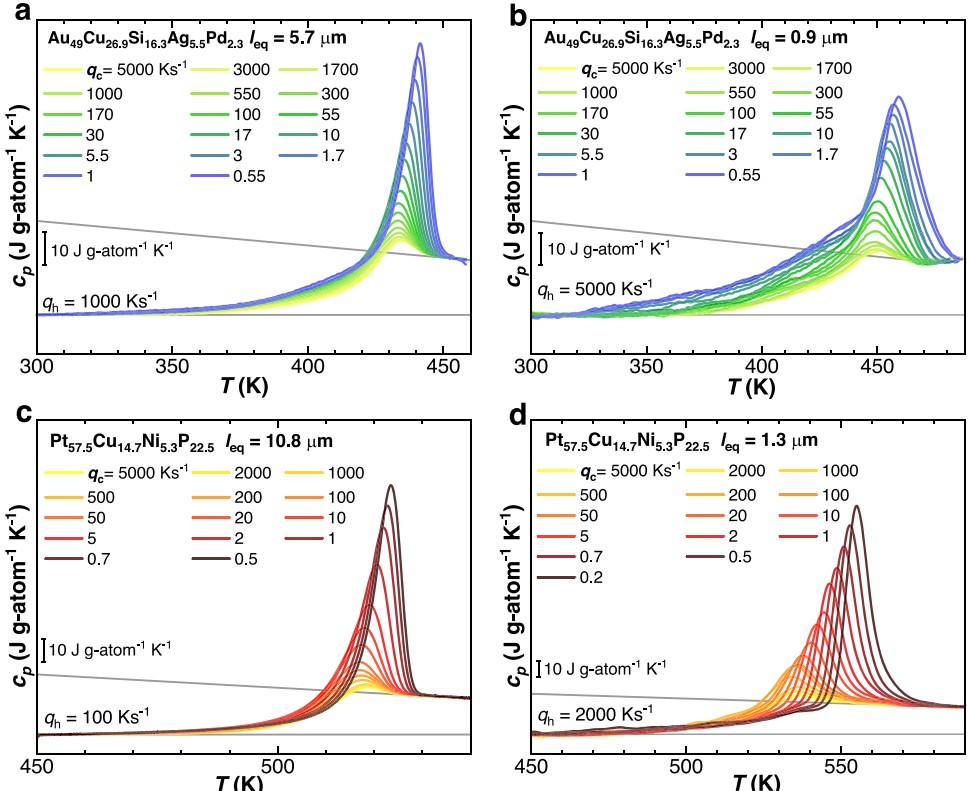

**Fig. 2 | Vitrification behavior of metallic glasses with different sizes.** Specific heat scans at heating rates, $q_h$, after cooling at the indicated rates, $q_c$, for $Au_{49}Cu_{26.9}Si_{16.3}Ag_{5.5}Pd_{2.3}$ at.% with $l_{eq} = 5.7\,\mu m$ (**a**) and 0.9 μm (**b**), as well as for $Pt_{57.5}Cu_{14.7}Ni_{5.3}P_{22.5}$ at.% with $l_{eq} = 10.8\,\mu m$ (**c**) and 1.3 μm (**d**). The gray lines are linear fits of the glass and the melt-specific heat and coincide with those reported in the inset of Fig. 1.

(see Fig. 1) generally applies to all frequencies, as indicated by the complete overlap of temperature-dependent $\tau$ at different sample sizes. All temperature-dependent $\tau$ data can be fitted by the empirical Vogel-Fulcher-Tammann (VFT) equation: $\tau = \tau_0 \exp(D^* T_0/(T - T_0))$, where $\tau_0$, $D^*$, and $T_0$ are the pre-exponential factor, the fragility index, and the Vogel temperature, respectively. The fitting parameters are indicated in the caption of Fig. 3 and are in accordance with previous studies[10,43].

The characterization of vitrification kinetics is presented in Fig. 2, showing specific heat scans obtained at the indicated heating rate, $q_h$, after cooling using a wide range of rates. In Fig. 2, we present the results obtained with two samples of different sizes for each alloy composition. However similar results, reported in Supplementary Fig. 2, were obtained for samples of other sizes. The very general feature observed in all panels of Fig. 2 is the expected development of a pronounced endothermic overshoot, in proximity of the step due to the glass transition. Its intensity grows with decreasing the previously applied cooling rate, signifying the attainment of lower enthalpic states in glasses cooled at lower rates[5]. Furthermore, we observe an additional kinetic phenomenon, that is, a low-temperature endothermic excess—governed by an underlying sub-$T_g$ relaxation—whose origin is discussed later. This is mostly visible in the Au-based glass former.

Figure 3a, b provides an overview of how different cooling rates convey the supercooled melt to glasses with different thermodynamic states in terms of $T_f$. Here, $T_f$ values were determined through the Moynihan method[44] (see Supplementary Note 1), with liquid, $c_p^l$, and glass, $c_p^g$, specific heats obtained from linear fits of the reversing specific heat in the liquid and glass regions, respectively (see insets of Fig. 1 and right panel of Supplementary Fig. 1), encompassing a wide range of temperature, which for the $c_p^l$ is as large as 60 K. These values are in general agreement with previously published data[43,45]. A

complementary representation is reported in Fig. 3c, which depicts a three-dimensional mapping on the way $T_f$ deviates from the bulk value at different cooling rates when changing sample size. For large cooling rates, vitrification takes place with the same dependence as that of the $\alpha$ relaxation time, independently of the sample size and in a bulk-like fashion. As a result, in this case, cooling rate-dependent $T_f$ can be fitted by the VFT equation with the same $D^*$ and $T_0$ values as those of the $\alpha$ relaxation. Hence, at large cooling rates, we can identify the main $\alpha$ relaxation as the leading mechanism of vitrification. However, decreasing the cooling rate entails marked deviation from the behavior expected if exclusively the $\alpha$ relaxation assisted vitrification (blue and red VFT-lines in Fig. 3a, b, respectively). The deviation is mild for the largest investigated samples, as previously reported for $Au_{49}Cu_{26.9}Si_{16.3}Ag_{5.5}Pd_{2.3}$ at.%[10]. In this case, the vitrification behavior is bulk-like, as no size effects on $T_f$ are detected on further increasing the sample size. In contrast, in this study we observe that reducing the sample size results in increasingly larger decoupling of vitrification kinetics from the $\alpha$ relaxation, indicating the increasingly prominent role of fast non-$\alpha$ mechanisms assisting vitrification at low rates. The $T_f$ reduction in the smallest samples with respect to that expected if only the $\alpha$ relaxation assisted vitrification is as large as >-40 K for both glasses for the lowest cooling rate at which we were able to vitrify the completely amorphous sample, that is, avoiding crystallization. Note that at lower rates we observed a decrease of the $c_p$ step at the glass transition indicating partial crystallization. In this case, the corresponding data were not used in Fig. 3. Interestingly, when the distance of $T_f$ from the bulk value is considered, as shown in Fig. 3c, the two investigated MG formers exhibit the same size and cooling rate dependence.

Both glasses exhibit similar behavior in terms of $T_f$ reduction. However, it is worth pointing out that a qualitative difference between the calorimetric response of these two glasses exists. In the

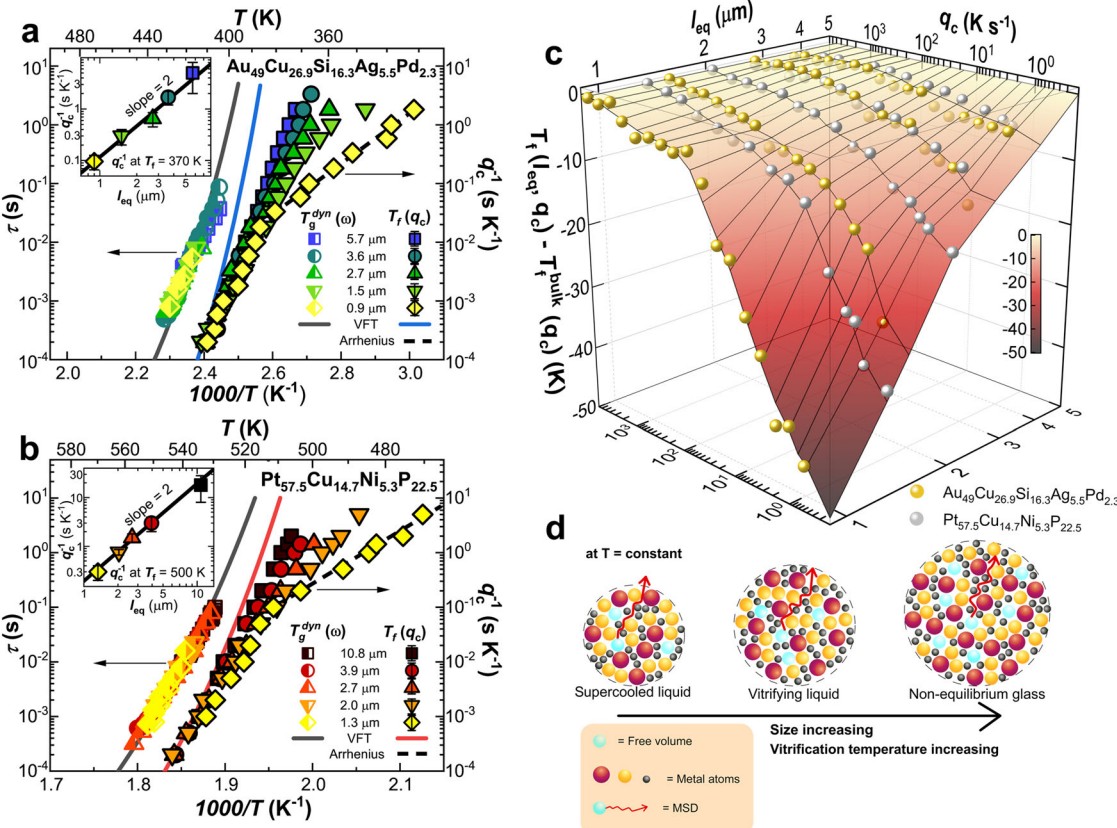

**Fig. 3 | Overview of temperature-dependent vitrification behavior and atomic mobility; and a sketch of FVHD model. a, b** Temperature dependence of the $\alpha$ relaxation time (left axis) and $T_f$ dependence on the inverse of cooling rate (right axis). Error bars are ±2 K in both $T_f$ and the temperature of a given $\tau$. The continuous gray; red and blue lines are VFT fits to $\tau$ and $q_c$ dependent $T_f$, respectively, with: $D^* = 9.8$; $T_0 = 311$ K for Au-based MG[43]; and $D^* = 7.4$; $T_0 = 426$ K for Pt-based MG. The dashed black lines are Arrhenius fits of vitrification kinetics in the low $T_f$ regime. The insets show the inverse of the cooling rate as a function of the equivalent length at the indicated $T_f$. **c** $T_f$ depression with respect to the $T_f$ of bulky samples as a function of $q_c$ and $l_{eq}$ for both investigated glasses. **d** Schematic representation of the FVHD model where, at a given temperature, a system can be, depending on its size, at equilibrium, vitrifying, or glassy.

$Au_{49}Cu_{26.9}Si_{16.3}Ag_{5.5}Pd_{2.3}$ at.% MG, the presence of non-$\alpha$ fast mechanisms of equilibration reflects on the presence of pronounced low-temperature excess endotherm, whose intensity increases with decreasing the cooling rate and the sample size (see Fig. 2, upper panels). An analogous phenomenology has been documented in a wide variety of glasses, including metals[46–48], plastic crystals[49], polymers[50–53], glucose[54], and phase change materials[55], mainly when aged for prolonged times at temperatures way below $T_g$. Its presence is intimately linked to the fast mechanism of relaxation assisting glass equilibration at low temperatures, where the $\alpha$ relaxation acquires experimentally unfeasible time scales. In the case of the $Pt_{57.5}Cu_{14.7}Ni_{5.3}P_{22.5}$ at.% glass former, the low-temperature excess endotherm can also be detected if the specific heat in excess to the high cooling rate reference, $c_{p,ex}$ is shown (see Supplementary Fig. 3). However, its intensity appears to be considerably smaller than in the case of the $Au_{49}Cu_{26.9}Si_{16.3}Ag_{5.5}Pd_{2.3}$ at.% glass former suggesting that this only accounts for a part of the fast mechanism of relaxation and most of the calorimetric signature of non-$\alpha$ mechanisms is convoluted with the $\alpha$ relaxation, a behavior analogous to that of glasses aged not too far below $T_g$[56,57].

## Discussion

The glass transition can be seen from the perspective of the characterization of the linear response in terms atomic/molecular motions caused by spontaneous fluctuations or that of unveiling the kinetic pathway transforming the supercooled liquid into a glass. Though related, these two aspects are conceptually different as the former

deals with the unperturbed glass at equilibrium, whereas vitrification entails the kinetic transformation resulting from a large non-linear perturbation, in this case, a temperature ramp[58]. Exploiting the capabilities of FSC permitted us to attain insights on both aspects of MG and their dependence on sample size. The sample size independence of atomic mobility associated with the $\alpha$ relaxation is expected considering that the length scale of the $\alpha$ relaxation never exceeds several nanometers[59,60]. This result is in line with those obtained in polymeric glass formers for sizes of several micrometers[34]. As a result of the limited size of the $\alpha$ relaxation, modifications of dynamics, if any[30], are observed for a sample size of the order of nanometers[31], which is by orders of magnitude smaller than the here considered sample sizes.

Vitrification kinetics, even for the largest samples, deviates from the behavior of the $\alpha$ relaxation, an aspect already evidenced and discussed in the $Au_{49}Cu_{26.9}Si_{16.3}Ag_{5.5}Pd_{2.3}$ at.% bulk glass former[10]. In this context, the role of equilibration mechanisms beyond the $\alpha$ relaxation has been demonstrated[10]. In the present work, we have shown that fast non-$\alpha$ equilibration mechanisms convey MG formers to increasingly smaller $T_f$ when the sample size is decreased. Importantly, this takes place at sample length scales in the micrometer range, an unprecedented result in MG, and is magnified at increasingly lower cooling rates. Furthermore, it is worth pointing out that, while vitrification at high cooling rates is size-independent and completely coupled to the $\alpha$ relaxation with VFT temperature dependence, size effects appear to be of importance where fast non-$\alpha$ equilibration mechanisms with mild temperature dependence play a major role in assisting vitrification.

For the very same reason that spontaneous fluctuations could be perturbed at length scales way shorter than the ones explored in the present study, the origin of the large $T_f$ depression observed in increasingly smaller samples but always larger than 1 μm must be sought on physical grounds where an additional size-dependent order parameter is introduced beyond the role of atomic mobility. In a completely general framework, the cooling rate-dependent fictive temperature varies as a result of temperature-dependent relaxation time—in this case of fast relaxation processes—and geometric factors, included in a function $g(l_{eq})$: $q_c^{-1} = g(l_{eq})\tau$, where $q_c$ is the applied cooling rate and $\tau$ the relaxation time. This is a generalization of the so-called Frenkel-Kobeko-Reiner relation connecting vitrification to dynamics[2,18], that is, the macroscopic kinetic transformation of a non-equilibrium system to spontaneous fluctuations in the system. The connection between the kinetic transformation taking place in glass aging and spontaneous fluctuations has been recently demonstrated[61]. In the following, we provide strong arguments indicating that the underlying physics behind the function $g(l_{eq})$ can be suitably captured by the free volume holes diffusion (FVHD) model.

The idea that glass equilibration can be assisted by diffusion of free volume holes towards a free interface and their removal to the outer world was already proposed by Alfrey et al.[62] and Curro et al.[63]. While diffusion is of no relevance for bulk glasses, recently, the FVHD model has been revitalized to account for accelerated glass equilibration in polymer films[64-67] and nanocomposites[68]. Diffusion of free volume holes was directly visualized in the devitrification of vapor-deposited colloidal glasses[36]. Within the FVHD model, time-dependent Fickian diffusion of free volume holes in the glass is described by:

$$\langle x^2 \rangle = 2Dt \qquad (1)$$

where $\langle x^2 \rangle$ is the mean squared displacement (MSD) and $D$ the diffusion coefficient. The latter is related to atomic motion in the glass and, therefore, is size-independent being directly related to $\tau$, via the (fractional) Stokes–Einstein and the Maxwell relations[69]. Here, it is worth pointing out that eq. (1) has been written for the case of one-dimensional diffusion, which, strictly speaking, is valid only for films. However, considering that the size of free volume holes is orders of magnitude smaller than the radius of curvature of spheres studied by us, the film approximation can be considered valid for all our samples.

Within a very general framework, vitrification starts to take place on cooling at a given rate, $q_c$, the farthest free volume holes from the free interface, located at $l_{eq}/2$ from the interface, is only able to displace at such interface without being expelled out of it. This scenario is schematically depicted in Fig. 3d, where the vitrifying system is presented in the middle sketch. The same panel shows how, in a system with smaller size (left sketch in Fig. 3d) at the same temperature, the MSD is large enough to maintain equilibrium. The opposite holds for a system with larger size (right sketch in Fig. 3d), where, at the same temperature, free volume holes are unable to diffuse out of the free interface, thereby making the system glassy. In the present study, we characterize vitrification in terms of $T_f$ that is approximately the mean value of the temperature range of vitrification[44]. In this range, the flux of free volume holes through the free interface crosses from the steady state value in the liquid to zero in the glass. As the flux of free volume holes depends on the amount of free interface, and therefore on the inverse of $l_{eq}$, samples with identical $l_{eq}$ will exhibit the same $T_f$ independently of the geometry. Hence, at the glass transition, eq. (1) can be re-written as:

$$l_{eq}^2 \sim 2D(T_f)q_c^{-1} \qquad (2)$$

or equivalently:

$$\log q_c^{-1} \sim 2\log l_{eq} - \log 2D(T_f) \qquad (3)$$

To test the validity of the FVHD model via eq. (3), we have considered the cooling rate providing a fixed $T_f$ at different $l_{eq}$ in the regime where fast non-$\alpha$ mechanisms of equilibration dominate. This was $T_f = 370$ and 500 K for $Au_{49}Cu_{26.9}Si_{16.3}Ag_{5.5}Pd_{2.3}$ at.% and $Pt_{57.5}Cu_{14.7}Ni_{5.3}P_{22.5}$ at.%, respectively. An identical outcome emerges choosing different $T_f$ values in the fast non-$\alpha$ mechanism regime. The result of this analysis is presented in the insets of Fig. 3a, b, where the logarithm of the inverse of the cooling rate is plotted as a function of the logarithm of $l_{eq}$. As can be observed, $q_c^{-1}$ varies with the square of $l_{eq}$, which perfectly fulfils the prediction of eq. (3), thereby validating the hypothesis of FVHD driven vitrification. Further corroboration emerges from the results of $Au_{49}Cu_{26.9}Si_{16.3}Ag_{5.5}Pd_{2.3}$ at.% samples with identical $l_{eq}$ but different geometry. The two samples exhibit identical cooling rate-dependent $T_f$ in agreement with the predictions of the FVHD model (see Supplementary Fig. 4).

The presence of non-$\alpha$ fast mechanisms of equilibration requires the search for the underlying atomic process. Apart from standard secondary relaxation processes[21,70,71], recent efforts have stressed the presence of liquid-like zones deep in the glassy state where the $\alpha$ relaxation is of no relevance[72]. These may be of importance in the vitrification process delaying lower temperatures the transformation from a liquid into glass with respect to expectations only accounting for the $\alpha$ relaxation. The structural relevance of liquid-like moieties is reflected in the presence of shear transformation zones (STZ)[73,74], where the presence of low energy barriers allows flow even deep in the glassy state. Liquid-like zones have been also identified by mechanical experiments showing the ability to relax substantial stress in MG, even deep in the glassy state[75]. Very recently, Napolitano and co-workers[22] have identified a slow Arrhenius process (SAP), bearing potential in the kinetics of equilibration of different phenomena in amorphous materials, including glass equilibration. They identified an intimate relation between SAP activation energy and the material's $T_g$. Data shown in Fig. 3a, b allows extracting the activation energy of the fast mechanisms of vitrification via the Arrhenius equation: $q_c^{-1} = q_{c,0}^{-1} \exp(E_a/kT)$, shown as dashed lines. This is done only for the smallest samples, for which the crossover from $\alpha$ to non-$\alpha$ controlled regime is fully attained. In this case, the data range to determine $E_a$ encompasses a temperature interval of >50 K, which makes the obtained values reliable and significant. For larger samples, the range of data where the Arrhenius fit can be reliably performed is either too limited or absent, due to the incipient crossover to the highly activated $\alpha$ relaxation regime. However, if data at lower cooling rates were available, this could be done for larger samples too. The resulting activation energies are: $E_a = 75 \pm 5$ kJ/mol and $E_a = 180 \pm 10$ kJ/mol for $Au_{49}Cu_{26.9}Si_{16.3}Ag_{5.5}Pd_{2.3}$ at.% and $Pt_{57.5}Cu_{14.7}Ni_{5.3}P_{22.5}$ at.%, respectively. These values are compatible with the expected activation energies of the SAP for the two glasses considering $T_g = 372$ K and $T_g = 505$ K for $Au_{49}Cu_{26.9}Si_{16.3}Ag_{5.5}Pd_{2.3}$ at.% and $Pt_{57.5}Cu_{14.7}Ni_{5.3}P_{22.5}$ at.%, respectively (see Fig. 3 of ref. [22]). This outcome makes the SAP a suitable candidate as the atomic mechanism assisting vitrification at low cooling rates. On more theoretical frameworks, the presence of different equilibration mechanisms can be derived on the base of the self-consistent Langevin equation[76] and the random first-order transition theory[77].

The size-dependent $T_f$ depression underlines the ability of small samples to maintain and reach equilibrium faster when subjected to a large non-linear stimulus, in this case, a temperature ramp. However, this can be extended to other types of stimuli, including mechanical stress. It has been observed that the propensity for shear localization in MGs decreases with decreasing sample size[78-83]. The structural disorder created by STZ can be effectively equilibrated in smaller samples resulting in more homogeneous-like deformation. Hence, the increasing ductility of MG when the sample size is reduced to microns[39] is naturally

## Au$_{49}$Cu$_{26.9}$Si$_{16.3}$Ag$_{5.5}$Pd$_{2.3}$ (UFS chip / Flash DSC 1)

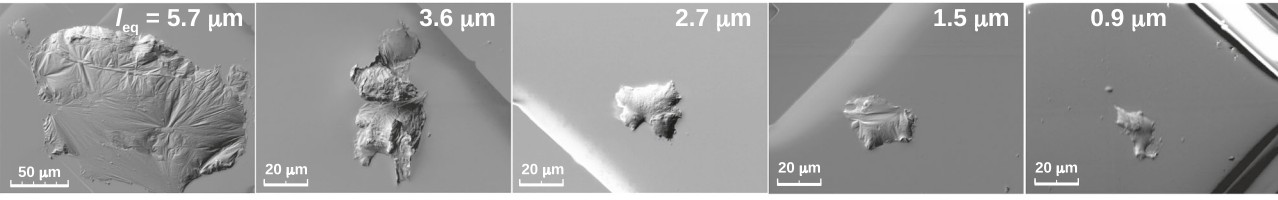

## Pt$_{57.5}$Cu$_{14.7}$Ni$_{5.3}$P$_{22.5}$ (HTS chip / Flash DSC 2+)

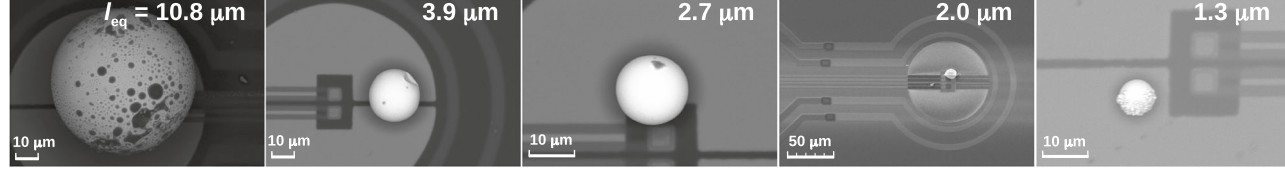

**Fig. 4 | Overview of SEM for all samples.** SEM micrographs of Au$_{49}$Cu$_{26.9}$Si$_{16.3}$Ag$_{5.5}$Pd$_{2.3}$ at.% (top row) and Pt$_{57.5}$Cu$_{14.7}$Ni$_{5.3}$P$_{22.5}$ at.% (bottom raw) glasses in order of characteristic length $l_{eq}$. The chip sensor and Flash DSC device used for the experiments are indicated.

explained by the ability of these small samples to promptly respond to a non-linear mechanical stimulus. Here, it is noteworthy that—despite the low $T_f$ attained, indicating strong thermodynamic stability—the ability to respond promptly to a non-linear external stimulus makes small samples kinetically unstable. Hence, the criterion on glass ductility exclusively based on the $T_f$ value[11,12] requires refinement to account for the glass size-dependent kinetic stability, that is, how this low $T_f$ has been achieved.

As a final important observation, it is worth of remark that the smallest samples of both investigated MG attain a thermodynamic state with $T_f$ 40 K lower than that of the corresponding bulk material at cooling rates of the order of 1 K s$^{-1}$, corresponding to observation time scales in the order of seconds. This underlines the attainment of remarkably low energies, which, in bulk glasses, would be reached only after prolonged natural aging[50,57,84,85]. Hence, one can expect that aging under appropriate conditions, where the observation time scales can be enlarged at wish significantly above a few seconds, can be used to create very low energy glassy states in micron-size samples, otherwise hardly attainable in bulk glasses, and bearing potential to convey insights on issues of extraordinary importance, such as the existence of the ideal glass[86–89].

## Methods

### Materials
We chose to study two bulk MG systems with very different compositions and, therefore, different chemical affinities among components, which points toward the universality of our findings. The gold-based MG, with composition Au$_{49}$Cu$_{26.9}$Si$_{16.3}$Ag$_{5.5}$Pd$_{2.3}$ at.% was produced as-spun ribbon of 7 ± 1 μm of thickness following a two-step procedure[43]. First, the mixture of the raw elements with purity 99.995% was melted and homogenized at ~1100 K in an alumina crucible in an Indutherm MC15 tilt-caster. The melt was tilt-cast into a water-cooled Cu-mold obtaining a rod of 5 mm in diameter and length of 34 mm. The rod was inductively remelted in a quartz tube and injected using a self-build melt spinner onto a rotating copper under argon atmosphere to obtain ribbons. The rod and the ribbons were proved to be XRD amorphous.

Nanoneedles of the Pt-based MG, with composition Pt$_{57.5}$Cu$_{14.7}$Ni$_{5.3}$P$_{22.5}$ at.%, were obtained according to the procedure developed in refs. 11,39 as detailed in the following. A crystalline master alloy was prepared by melting the high-purity elements in a vacuum-sealed quartz tube at 1300 K. The alloy was remelted in a thin quartz tube under argon followed by water quenching, which resulted in the formation of glassy rod with 2 mm diameter and 60 mm length. The Pt-based MG nanofibers were prepared by thermoplastic press-

and-pull technique. A small piece (~20 mg) of Pt-based MG was pressed under 700 N at 543 K against a steel mesh consisting of 200 μm diameter cylindrical cavities. After pressing, the MG was pulled away from the steel mesh at a speed of 20 mm/min while maintaining the temperature at 543 K. The thermoplastic press-and-pull technique produced long Pt-based MG nanofibers attached to the steel mesh. The press-and-pull experiments were conducted using a custom-built heating platen setup attached to a universal mechanical testing machine. Short sections were mechanically clipped from the MG nanofibers and harvested for calorimetric analysis.

### Scanning electron microscopy
For FSC measurements, five specimens of different sizes were obtained by manually cutting either an Au-based ribbon or a Pt-based needle under an optical microscope. Cutting from the tip of the nanoneedles facilitated the procedure of obtaining smaller and smaller samples for the Pt-based glass. The masses of the samples ranged from 3 to 2000 ng.

The morphology of the sample was assessed by scanning electron microscopy (SEM) performed directly on the FSC chips. We used a Hitachi TM3000 Tabletop Microscope for the Au-based samples and a Zeiss Sigma VP (secondary electron mode) for the Pt-based samples. We determined the characteristic lengths of the Flash DSC specimens, in terms of the ratio between sample volume and free surface area. This metric is independent of the sample geometry and is the relevant parameter for the application of the FVHD model. SEM micrographs of all of the used specimens are shown in Fig. 4. The observed sample geometry was naturally obtained allowing the material to flow repeatedly heating the sample on the chip-sensor prior to the SEM analysis above the melting temperature. The sample geometry highly depends on the composition of the specimen and its wetting behavior with the substrate underneath.

The mass and geometrical parameters of each specimen are listed in Supplementary Table 1 and they are determined as follows. We first estimated the mass of the sample from the heat flow rate step, $\Delta(HF)$, at a defined temperature in proximity of the glass transition, i.e., 420 K and 520 K for Au and Pt-based MG, respectively. From the knowledge of the specific heat, $c_p$, from conventional calorimetry, we determined the sample mass as $m = \Delta(HF)/(c_p q_H)$, where $q_H$ is the heating rate of the experiment. The volume of the specimen, $V$, was determined from the ratio of the estimated mass of the specimen and the calculated density of the MG.

The sample-free surface, $A$, was estimated in two different ways, depending on the geometry of the sample. For the Au-based specimens

molten on ultra-fast sensors (UFS) and employed in the Flash DSC 1, a film-like geometry was obtained, revealing a complete wetting behavior of this alloy with the substrate of the active area of this sensor, made of aluminum. The sample-free surface was determined from SEM micrographs considering only the upper area (the lower being buried by the substrate). This area was determined using the dedicated freeware GIMP2.10 which allows tracing the sample perimeter. Subsequently, the program allows assessing the sample surface inscribed in the perimeter by partitioning the image into pixels areas. The Pt-based specimens were deposited on high-temperature sensors (HTS), suitable for the Flash DSC 2+ analyses. The active area of these sensors consists of a membrane made of silicon nitride. In this case, the wetting of the liquid specimen with the substrate was hindered and a spherical geometry was obtained. In this case, the surface area and the sample volume were estimated by measuring the diameter of the sphere. Given the well-defined geometry detected by SEM in spherical samples, a dedicated program for morphology analysis was not required. As said above, the $Au_{49}Cu_{26.9}Si_{16.3}Ag_{5.5}Pd_{2.3}$ at.% composition was mostly characterized using the Flash DSC 1 and UFS sensors, whereas the $Pt_{57.5}Cu_{14.7}Ni_{5.3}P_{22.5}$ at.% composition using the Flash DSC 2+ using HTS sensors. However, one sample of $Au_{49}Cu_{26.9}Si_{16.3}Ag_{5.5}Pd_{2.3}$ at. % was also characterized in the Flash DSC 2+ using a HTS sensor. This specimen after deposition and melting on the HTS sensor did not show complete wetting behavior and resulted in a sphere-like specimen bearing an irregular shape.

### Calorimetric characterization

Different fast scanning devices and sensors were used for calorimetric experiments. Flash DSC 1, operating between 173 and 723 K was used to characterize the $Au_{49}Cu_{26.9}Si_{16.3}Ag_{5.5}Pd_{2.3}$ at.% glass former. One sample of this glass former is also investigated using the Flash DSC 2+ (see Supplementary Fig. 4 for details). The $Pt_{57.5}Cu_{14.7}Ni_{5.3}P_{22.5}$ at.% glass former is analyzed exclusively with the flash DSC 2+ with a maximum operating temperature of 973 K. The samples were inserted manually into the active area of the sensors using an optical microscope and a single brush hair. Flash DSC 1 was purged with nitrogen, whereas Flash DSC 2+ with both nitrogen and argon gas at a flow rate of 20 ml/min. Temperature calibration was performed with indium standard deposited on the reference area. Prior to experiments, samples were stabilized onto the sensor by means of a standard pre-treatment that included melting at 723 K for $Au_{49}Cu_{26.9}Si_{16.3}Ag_{5.5}Pd_{2.3}$ at.% or 973 K for $Pt_{57.5}Cu_{14.7}Ni_{5.3}P_{22.5}$ at.%, that is, above the alloys melting temperatures, i.e., $T_m^{Au49} = 673$ K and $T_m^{Pt57} = 873$ K, respectively. After the melting step, it followed a rapid cooling with constant cooling rate $q_c = 5000$ K s⁻¹ down to room temperature.

The kinetics of vitrification was assessed at cooling rates between 0.55 and 3000 K s⁻¹ for $Au_{49}Cu_{26.9}Si_{16.3}Ag_{5.5}Pd_{2.3}$ at.% and between 0.2 and 5000 K⁻¹ for $Pt_{57.5}Cu_{14.7}Ni_{5.3}P_{22.5}$ at. %. To avoid crystallization, samples were melted above the liquidus temperature and quenched at 5000 K s⁻¹ down to 443 K for $Au_{49}Cu_{26.9}Si_{16.3}Ag_{5.5}Pd_{2.3}$ at.% and 553 K for $Pt_{57.5}Cu_{14.7}Ni_{5.3}P_{22.5}$ at.%. Subsequently, variable cooling rates were applied down to 183 K. The heating scan was applied with heating rates between 100 and 5000 K s⁻¹, depending on the sample mass (see Supplementary Table 1). The optimal heating rate for each sample has been chosen as a compromise between maximizing the signal-to-noise ratio and minimizing the thermal lag. This procedure required running identical trial tests on a few samples of different size. The fictive temperature, $T_f$, was calculated via the Moynihan method[44] (see Supplementary Fig. 1 for details).

Step-response analyses, consisting of up-jumps followed by isotherms, from glassy to the liquid state, was used to assess the atomic mobility[40,41]. Two main step protocols were used, a first consisted of an up-jump of 2 K with a nominal heating rate of 2000 K s⁻¹ followed by a 0.05 s isotherm, that is, a base frequency of 20 Hz, and a second with up-jumps of 2 K at 200 K s⁻¹ and 1 s isotherms to assess frequency response with a base frequency of 1 Hz. The frequency-dependent complex specific heat, $c_p^*(\omega)$, was calculated by Sliding Fast Fourier Transformation of the heat flow rate and instantaneous heating rate:

$$c_p^*(\omega) = \frac{\int_0^{t_p} HF(t)e^{-i\omega t}dt}{\int_0^{t_p} q_h(t)e^{-i\omega t}dt} \tag{4}$$

which was repeated for each period of oscillation $t_p$. Accessing higher harmonics allows for assessing the complex specific heat frequency response from 1 to 150 Hz for Flash DSC 1 or up to 500 Hz in the case of Flash DSC 2+. The temperature-dependent relaxation time, $\tau = 2\pi/\omega$ is determined at the inflection point of $c_p'$. It is worth pointing out that, in the case of nominal heating rate of 2000 K s⁻¹, due to the small temperature step and thermal inertia, the actual heating rate was systematically smaller than the nomimal one. However, the heat flow rate is strictly correlated to the instantaneous heating rate, thereby delivering accurate determination of $c_p^*$.

## Data availability

The authors declare that the data supporting the findings of this study are available within the article and its Supplementary Information files. These as raw data are available from the corresponding authors upon request.

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

## Acknowledgements

The authors would like to thank Maryam Rahimi Chegeni for support with some calorimetry work and Giulia Ferrero for designing Fig. 3d. This work has received funding from the projects PID2021-123438NB-I00 funded by MCIN/AEI/10.13039/501100011033 and by "ERDF A way of making Europe" (D.C.); PID2020-114506GB-I00 funded by the Spanish Minister of Science and Innovation (D.C.); Eusko Jaurlaritza, code: IT-1566-22 funded by The Basque Government (D.C.); TED2021-129457B-I00 funded by the Spanish Minister of Science and Innovation (D.C.); US National Science Foundation through CAREER Award #1921435 (G.K.); the Office of Naval Research under grant N00014-20-1-2200 (J.S.); the German Federation of Industrial Research Associations through grant IGF 21469N (R.B., I.G., and N.N.); and the German Research Foundation through grant DGF GA1721/4-1 (R.B., I.G., and N.N.).

## Author contributions

D.C., I.G., V.D.L. conceived the idea and designed the experiments. I.G. and G.K. produced the samples. V.D.L., S.S.R., N.N., and M.F. carried out the calorimetric survey and the SEM analysis. D.C., I.G., and V.D.L. analyzed and interpreted the results. D.C. wrote the original manuscript with the help of I.G. and V.D.L. All authors contributed to interpreting the results and to the revision of the manuscript.

## Competing interests

The authors declare no competing interests.
