## [Peer Review File · Nature Communications]

Size dependent vitrification in metallic glassesREVIEWER COMMENTS

Reviewer #1 (Remarks to the Author):

The size dependent vitrification in Au and Pt-based metallic glasses was investigated in this work. The novel method was used to reduce the length scale of metallic glasses. Their vitrification behavior and atomic mobility were studied. The main α relaxation process remains size independent, ranging from bulk to several microns. However, the pronounced size dependent vitrification kinetics is observed for the smallest samples and at low cooling rates, which result in more than 40 K decrease in fictive temperature compared with bulk samples.

The work presents good scientific quality and the results shown are original and interesting, there are some points the author should refer to before the final decision on publication. The specific issues are described below.

1. The Au₄₉Cu_{26.9}Si_{16.3}Ag_{5.5}Pd_{2.3} and Pt_{57.5}Cu_{14.7}Ni_{5.3}P_{22.5} metallic glasses are selected in this work. However, what's the reason that these two compositions are chosen, and the link between Au and Pt-based MGs. The reviewer concerns that the author normally select the same chemical elements or the materials show the same β relaxation or other important characteristics.

2. The author shows very interesting and original method to calculate the mass, volume, free surface, and equivalent size. It is better to define the equivalent size in the results part to show clearly the size dependent since it may mislead the readers that the size effect just the same as macroscopic level or down to nanometers.

3. In figures 3 (a)-(b), the temperature dependent vitrification behavior and characteristic time are shown. One of them fitted with VFT while the other one fitted with Arrhenius. The reviewer is interested with the link between the vitrification behavior and atomic mobility.

4. The author claims that $D^*=9.8$ of Au-based MG while $D^*=7.4$ of Pt-based MG. The relevant reference [Y. J. Duan, L. T. Zhang, J. C. Qiao, Y. J. Wang, Y. Yang, T. Wada, H. Kato, J. M. Pelletier, E. Pineda, and D. Crespo, Intrinsic Correlation between the Fraction of Liquidlike Zones and the β Relaxation in High-Entropy Metallic Glasses, *Phys. Rev. Lett.* 129, 249902 (2022).] shows the difference of fragility between the samples with different β relaxation and the high-entropy MGs vs 'conventional' MGs. Therefore, what is the reason of different behavior of Au and Pt-based MGs.

In summary, the size dependent vitrification in Au and Pt-based metallic glasses revealed in this manuscript is interesting and original. The work is interesting but the manuscript is not ready for publication in its present form and some important issues require clarification.

Reviewer #2 (Remarks to the Author):

This paper presents a thorough and inspiring analysis on Au and Pt-based MGs using fast scanning calorimetry where the vitrification kinetics in a range of time scales were analyzed. In that respect, it is the first time authors observe pronounced size dependent vitrification kinetics more evident for the

smallest samples and at low cooling rates. Furthermore authors show linear response in terms of thermal susceptibility for metallic glasses of different size. Diffusion of free volume holes theory was proposed for the selected metallic glasses to bring insight to relaxation kinetics. This paper can be very helpful to the generation of thermodynamically ultra-stable metallic glasses; which is one of the current bottlenecks. For these reasons I am pleased to accept the paper in its current form.

Reviewer #3 (Remarks to the Author):

The authors investigated two metallic glass formers, an Au-based and a Pt-based alloy, by means of extremely fast scanning calorimetry to assess their relaxation behaviour. They followed their approach relatively recently published [10] for same the Au-based alloy as used in the current work. Both the atomic mobility and the vitrification kinetics were assessed by means of a stepwise temperature protocol and measurement of the fictive temperature, respectively. While the authors concluded in their previous work [10] that there is no size effect for the masses used (500 ng – 1000 ng), they now deliberately varied the masses of both glass-forming alloys in order to reveal possible size effects on the relaxation behaviour. In agreement with their published data, the alpha-relaxations seem to decouple from the vitrification kinetics. More importantly, and that is the main novelty of the present manuscript, they report that the decoupling depends on the sample size. They explain this phenomenon with a simple picture/model, viz. the free volume holes diffusion (FVHD) model. In doing so, a characteristic length scale is introduced, which the authors define as the volume-to-surface ratio of the sample. A smaller characteristic length scale is indicative of shorter diffusion paths required for the free volume holes to be annihilated at the surface. In combination with the typical relaxation time, the diffusion length decides whether or not a glass/supercooled liquid can attain metastable equilibrium, thereby governing the observed glass transition temperature. By reducing the sample size, lower fictive temperatures can be reached, or, equivalently, glassy states with a lower enthalpy can be obtained on slow cooling. Slow Arrhenius processes are suggested to be responsible for the relatively fast relaxation processes when the supercooled liquid is subjected to comparatively low cooling rates. Using their data, a value for the activation energy is extracted.

General comments:

The paper reads well and all statements as well as conclusions are explained and discussed in a comprehensible manner. The results are based on a very systematic and comprehensive experimental approach.

Specific comments:

1. I am a bit sceptical about the characteristic length scale and its experimental determination. Therefore, I would be grateful if you could provide more information on how exactly you measured. Did you use a dedicated programme as it is for example used for microstructural analysis? Did you just consider the top surface or two times the top surface?
2. You provide data for the Au-based, which was recorded in the DSC 1 and the DSC 2+ (SFig. 4). Owing

to the different chip materials, the wetting and contacting of the sample is rather different. You show that the different contact with the chip is not a factor and that the data obtained for a sample with nominally identical characteristic length scales are similar. I would have expected that the heat flow might be affected by the shape. Do you have any explanation for your observation?

Moreover, I wonder, if your FVHD model complies with SFig. 4a: if I am not mistaken your argument is that the supercooled liquid falls out of metastable equilibrium on cooling, once the "holes" cannot diffuse to the sample surface in the time given. The distance from the centre of the more spherical sample should be longer than in the spread-out sample, why it is surprising to me that the relaxation behaviour is identical.

3. The linear fit of c_p , melt (cf. Fig. 2) is crucial for the determination of the fictive temperature (see also SFig. 1). What is typical error made here, or, in other words, what is the typical error of T_f ? It would be really helpful if you could extend the temperature regime in Fig. 2 a and b to indicate the quality of your linear fit. Especially for the Au-based alloy heated at 5000 K/s the linear fit appears to be somewhat random -considering only the temperature window given in Fig. 2b.

4. If I am not mistaken you have used different heating rates for the determination of the fictive temperature (because of different sample masses). But for the step protocols always the same heating rate of 2000 K/s prior to the anneal has been applied? Is it because the isothermal step guarantees temperature equilibration?

5. Most of the figures -especially the legends and additional insets- are extremely small and difficult to read.

6. It would be probably helpful if you added the applied heating rates (used in the T_f -experiments) in Table 1 of the SI for the different samples.

7. Representation of Fig 3c somewhat difficult to read because it is not possible to unambiguously allocate the position of the data points in space. It would be helpful if lines of selected constant cooling rates could connect the corresponding data points. Why are the samples with largest characteristic length not included?

8. When I compare your current data depicted Fig. 3 a with Fig. 3 from [10] (or to be more precise with Fig. SM3 [10]), the T_f data of your present data seems to have a different curvature, which should result in different parameters of the VFT fit. How do the present data compare with your previous data?

9. You have only fitted the data of the smallest samples (for the Au-based as well as for the Pt-based glasses) with an Arrhenius equation. What is the physical explanation that this approach is valid for this sample ($l_{eq} = 0.9 \mu\text{m}$ or $1.3 \mu\text{m}$) but not for the second smallest samples? An Arrhenius fit would be also possible here and the slope would be completely different. Please provide a motivation for your approach because otherwise the calculated activation energy might be somewhat arbitrary.

Reviewer #1: The size dependent vitrification in Au and Pt-based metallic glasses was investigated in this work. The novel method was used to reduce the length scale of metallic glasses. Their vitrification behavior and atomic mobility were studied. The main alpha relaxation process remains size independent, ranging from bulk to several microns. However, the pronounced size dependent vitrification kinetics is observed for the smallest samples and at low cooling rates, which result in more than 40 K decrease in fictive temperature compared with bulk samples.

The work presents good scientific quality and the results shown are original and interesting, there are some points the author should refer to before the final decision on publication. The specific issues are described below.

Response: We thanks the Reviewer for acknowledging the relevance and quality of our work. In the following, we respond to all points raised.

1. The Au₄₉Cu_{26.9}Si_{16.3}Ag_{5.5}Pd_{2.3} and Pt_{57.5}Cu_{14.7}Ni_{5.3}P_{22.5} metallic glasses are selected in this work. However, what's the reason that these two compositions are chosen, and the link between Au and Pt-based MGs. The reviewer concerns that the author normally select the same chemical elements or the materials show the same α relaxation or other important characteristics.

Response: The reason of our choice stands precisely in selecting metal alloys with very different chemical composition and affinity among atomic components. In this way, we demonstrate that deviations from bulk behavior do not depend on the specific choice of the elements forming the metal alloy, thereby proving a more universal validity of our findings. While the two alloys exhibit different alpha relaxation behavior, the latter is shown to be size dependent, marking the analogy between the two alloys. Furthermore, Au-based and Pt-based metallic glasses are weakly prone to oxidation, which allows chemical stability during measurements. We have explained the reason of our choice adding the following text:

"We chose these two alloys because of their very different composition and, therefore, chemical affinity among components, which points toward the universality of our findings."

2. The author shows very interesting and original method to calculate the mass, volume, free surface, and equivalent size. It is better to define the equivalent size in the results part to show clearly the size dependent since it may mislead the readers that the size effect just the same as macroscopic level or down to nanometers.

Response: We agree with the Reviewer that the determination of the mass, volume, free surface and eventually equivalent size requires more emphasis. Hence, we have added the following text in the Results section:

"Here, as detailed in the Methods section, V and A were obtained from the diameter in spherical samples. In the case of film-like samples, while A was visually determined from SEM micrographs, V was obtained from the knowledge of the sample mass via the alloy density"

3. In figures 3 (a)-(b), the temperature dependent vitrification behavior and characteristic time are shown. One of them fitted with VFT while the other one fitted with Arrhenius. The reviewer is interested with the link between the vitrification behavior and atomic mobility.

Response: In line with the Reviewer comment, we have clearly specified that, at high cooling rates, vitrification is mediated by the main alpha relaxation and is size independent. Regarding the atomic mobility mediating vitrification behavior at low cooling rates, we have extensively discussed the lack of relation with the alpha relaxation in the Results section and, in the Discussion section, the potential alternative atomic mechanisms involved (page ..., lines ...). However, we notice that, in point 4, the Reviewer has suggested an interesting work on the presence of liquid-like zones deep in the glassy state. This aspect has been added in the discussion on the link between vitrification behavior and atomic mobility. In the Result and Discussion sections, we have added the following text:

“Hence, at large cooling rates, we can identify the main α relaxation as the leading mechanism of vitrification.”

and

“Liquid-like zones have been also identified by mechanical experiments showing the ability of relaxing substantial stress in metallic glasses, even deep in the glassy state⁷⁴.”

4. The author claims that $D^*=9.8$ of Au-based MG while $D^*=7.4$ of Pt-based MG. The relevant reference [Y. J. Duan, L. T. Zhang, J. C. Qiao, Y. J. Wang, Y. Yang, T. Wada, H. Kato, J. M. Pelletier, E. Pineda, and D. Crespo, *Intrinsic Correlation between the Fraction of Liquidlike Zones and the β Relaxation in High-Entropy Metallic Glasses*, *Phys. Rev. Lett.* 129, 249902 (2022).] shows the difference of fragility between the samples with different β relaxation and the high-entropy MGs vs ‘conventional’ MGs. Therefore, what is the reason of different behavior of Au and Pt-based MGs.

Response: As noticed by the Reviewer, both metallic glasses investigated in our study are conventional ones. However, their difference in fragility lays in the range of fluctuations for conventional metallic glasses (see e.g. *J. Non-Cryst. Sol.* 1995, 185 199-202) and, differently from high entropy alloys, they can be classified as rather fragile metallic glasses. Though the difference in fragility in metallic glasses may be significant, with D^* varying between 5 and 27, the slight fragility difference can be connected to their T_g difference. Indeed, for a given class of glasses, T_g and fragility are to some extent correlated.

In summary, the size dependent vitrification in Au and Pt-based metallic glasses revealed in this manuscript is interesting and original. The work is interesting but the manuscript is not ready for publication in its present form and some important issues require clarification.

Response: We thank again the Reviewer and are confident that our response and modifications to the manuscript are enough to make our manuscript acceptable for publication.

Reviewer #2 (Remarks to the Author):

This paper presents a thorough and inspiring analysis on Au and Pt-based MGs using fast scanning calorimetry where the vitrification kinetics in a range of time scales were analyzed. In that respect, it is the first time authors observe pronounced size dependent vitrification kinetics more evident for the smallest samples and at low cooling rates. Furthermore authors show linear response in terms of thermal susceptibility for metallic glasses of different size. Diffusion of free volume holes theory was proposed for the selected metallic glasses to bring insight to relaxation kinetics. This paper can be very helpful to the generation of thermodynamically ultra-stable metallic glasses; which is one of the current bottlenecks. For these reasons I am pleased to accept the paper in its current form.

Response: The Reviewer has completely caught the main message conveyed by our manuscript and the deep implications of our findings. We thanks him/her for the highly positive assessment of our manuscript.

Reviewer #3 (Remarks to the Author):

The authors investigated two metallic glass formers, an Au-based and a Pt-based alloy, by means of extremely fast scanning calorimetry to assess their relaxation behaviour. They followed their approach relatively recently published [10] for same the Au-based alloy as used in the current work. Both the atomic mobility and the vitrification kinetics were assessed by means of a stepwise temperature protocol and measurement of the fictive temperature, respectively. While the authors concluded in their previous work [10] that there is no size effect for the masses used (500 ng – 1000 ng), they now deliberately varied the masses of both glass-forming alloys in order to reveal possible size effects on the relaxation behaviour. In agreement with their published data, the alpha-relaxations seem to decouple from the vitrification kinetics. More importantly, and that is the main novelty of the present manuscript, they report that the decoupling depends on the sample size. They explain this phenomenon with a simple picture/model, viz. the free volume holes diffusion (FVHD) model. In doing so, a characteristic length scale is introduced, which the authors define as the volume-to-surface ratio of the sample. A smaller characteristic length scale is indicative of shorter diffusion paths required for the free volumes holes to be annihilated at the surface. In combination with the typical relaxation time, the diffusion length decides whether or not a glass/supercooled liquid can attain metastable equilibrium, thereby governing the observed glass transition temperature. By reducing the sample size, lower fictive temperatures can be reached, or, equivalently, glassy states with a lower enthalpy can be obtained on slow cooling. Slow Arrhenius processes are suggested to be responsible for the relatively fast relaxation processes when the supercooled liquid is subjected to comparatively low cooling rates. Using their data, a value for the activation energy is extracted.

Response: We thank the Reviewer for the detailed evaluation, which nicely describes the main objective, findings and implications of our study. In the following, we reply point by point to all concerns raised.

General comments:

The paper reads well and all statements as well as conclusions are explained and discussed in a comprehensible manner. The results are based on a very systematic and comprehensive experimental approach.

Specific comments:

1. I am a bit sceptical about the characteristic length scale and its experimental determination. Therefore, I would be grateful if you could provide more information on how exactly you measured. Did you use a dedicated programme as it is for example used for microstructural analysis? Did you just consider the top surface or two times the top surface?

Response: We agree with the Reviewer that the description of the determination of the characteristic length scale lacks some important information. While for spherical samples, such determination is straightforward, in the original version of our manuscript we did not describe in details how the upper surface area, relevant for diffusion of free volume holes, was determined. We have introduced this information in the Methods section:

"[The sample free surface was determined from SEM micrographs] considering only the upper area (the lower being buried by the substrate). This area was determined using the dedicated freeware GIMP2.10 that allows tracing the sample perimeter. Subsequently, the program allows assessing the sample surface inscribed in the perimeter by partitioning the image in pixels areas."

2. You provide data for the Au-based, which was recorded in the DSC 1 and the DSC 2+ (SFig. 4). Owing to the different chip materials, the wetting and contacting of the sample is rather different. You show that the different contact with the chip is not a factor and that the data obtained for a sample with nominally identical characteristic length scales are similar. I would have expected that the heat flow might be affected by the shape. Do you have any explanation for your observation?

Response: The Reviewer correctly points out the different thermal contact between sample and chip in Flash DSC1 and DSC2+. As noticed by the Reviewer, this may result in different heat flow rates when heating the sample. In our experiments, to minimize thermal lag, we systematically used heating rates for spherical sample lower than those applied to film-like samples. We have added a new panel in Supplementary Figure 4 (panel c) to address this point. Here, we show that the spherical sample, heated at 300 K/s, exhibits devitrification at lower temperatures than the film-like sample, heated at 2000 K/s. However, importantly, in the same panel we show that the area underlined below heat flow rate scans, reporting on the glass enthalpic state, is equal for the two samples, implying identical fictive temperatures. Altogether, this observation indicates that, depending on sample geometry and applied heating rate, devitrification on heating may be different as in this case, but the resulting fictive temperature is identical in samples with

equal characteristic length scales. In line with this response, we have added the following caption to describe panel c of Supplementary Figure 4:

“Comparison of heat flow rate scans after cooling at two rates of samples with identical size and film-like sample (measured by Flash DSC 1) and the irregularly shaped sphere sample (measured by Flash DSC 2+).”

and the following text in Supplementary Note 2

“resulting in different heat flow rate scans (see panel (c) of Supplementary Figure 4)”

Moreover, I wonder, if your FVHD model complies with SFig. 4a: if I am not mistaken your argument is that the supercooled liquid falls out of metastable equilibrium on cooling, once the “holes” cannot diffuse to the sample surface in the time given. The distance from the centre of the more spherical sample should be longer than in the spread-out sample, why it is surprising to me that the relaxation behaviour is identical.

Response: This is another important point raised by the Reviewer requiring careful consideration. In the manuscript, we discuss on the onset of vitrification as taking place when the most distant free volume hole from the free interface cannot diffuse out of the sample. However, in our study, we focus on the fictive temperature, which is rather representative of the mean temperature of the whole vitrification process. This depends on the flux of free volume holes through free interfaces. According to this argument, glasses with identical free interface, and therefore identical characteristic lengths, will exhibit the same T_f . We have introduced this argument adding the following text:

“In the present study, we characterize vitrification in terms of T_f that is approximately the mean value of the temperature range of vitrification. In this range, the flux of free volume holes through the free interface crosses from the steady state value in the liquid state to zero in the glass. As the flux of free volume holes depends on the amount of free interface, and therefore on the inverse of A_{eq} , samples with identical A_{eq} will exhibit the same T_f independently of the geometry.”

3. The linear fit of c_p , melt (cf. Fig. 2) is crucial for the determination of the fictive temperature (see also SFig. 1). What is typical error made here, or, in other words, what is the typical error of T_f ? It would be really helpful if you could extend the temperature regime in Fig. 2 a and b to indicate the quality of your linear fit. Especially for the Au-based alloy heated at 5000 K/s the linear fit appears to be somewhat random -considering only the temperature window given in Fig. 2b.

Response: Reading this comment we have realized that we did not clearly specified how we determine $c_{p,melt}$ (and also $c_{p,glass}$). This is done using reversing specific heats at different frequencies (insets of Figure 1), where, thanks to the absence of non-reversing contributions, the temperature range where $c_{p,melt}$ can be fitted is large enough to allow obtaining reliable T_f . To clarify this point, the following sentences have been added in the caption of Figure 1:

“The gray lines are linear fits to glass and liquid specific heats.”

and in the main text:

“Here, T_f values were determined through the Moynihan method\cite{Moynihan1976} (see Supplementary Note 1), with liquid, c_p^l , and glass, c_p^g , specific heats obtained from linear fits of the reversing specific heat in the liquid and glass regions, respectively (see inset of Fig. \ref{Fig.1}).”

4. If I am not mistaken you have used different heating rates for the determination of the fictive temperature (because of different sample masses). But for the step protocols always the same heating rate of 2000 K/s prior to the anneal has been applied? Is it because the isothermal step guarantees temperature equilibration?

Response: The set point value of 2000 K/s is the nominal heating rate applied to the sample. In reality, depending on the sample mass, the actual heating rate may vary from few hundreds K/s for the largest samples to more than 1000 K/s for the smallest. However, the heat flow rate is correlated with the instantaneous heating rate. Hence, the complex specific heat resulting from the ratio of Fourier transform of the heat flow rate and the heating rate (equation 4 of the main manuscript) does not depend on the deviation from the nominal heating rate. Corroborating these arguments, it was shown that using very different sample masses, resulting in a wide range of instantaneous heating rates, does not results in differences in the complex specific heat; see ref. 40 of the manuscript. To clarify this point, we have added the following text:

“It is worth pointing out that, in the case of nominal heating rate of 2000 K/s, due to the small temperature step and thermal inertia, the actual heating rate was systematically smaller than the nominal one. However, the heat flow rate is strictly correlated to the instantaneous heating rate, thereby delivering accurate determination of c_p^ .”*

5. Most of the figures -especially the legends and additional insets- are extremely small and difficult to read.

Response: In line with this comment, we have increase the size of legends throughout all figures of the manuscript.

6. It would be probably helpful if you added the applied heating rates (used in the T_f -experiments) in Table 1 of the SI for the different samples.

Response: As suggested, we have added the heating rates in Table 1 of the SI.

7. Representation of Fig 3c somewhat difficult to read because it is not possible to unambiguously allocate the position of the data points in space. It would be helpful if lines of selected constant cooling rates could connect the corresponding data points. Why are the samples with largest characteristic length not included?

Response: We have constructed a network of connections to help reading Figure 3c. The samples with largest characteristic length exhibit bulk-like behavior. Hence, they are taken as references to determine the T_f deviation from bulk behavior.

8. When I compare your current data depicted Fig. 3 a with Fig. 3 form [10] (or to be more precise with Fig. SM3 [10]), the T_f data of your present data seems to have a different curvature, which should result in different parameters of the VFT fit. How do the present data compare with your previous data?

Response: The curvature is practically equal; see Figure below, where data from ref. 10 are shown as yellow stars. The only difference is a slight vertical shift in T_f as a function of cooling rate, likely due to a different wetting behavior in the chip used in ref. 10.

9. You have only fitted the data of the smallest samples (for the Au-based as well as for the Pt-based glasses) with an Arrhenius equation. What is the physical explanation that this approach is valid for this sample ($l_{eq} = 0.9 \mu m$ or $1.3 \mu m$) but not for the second smallest samples? An Arrhenius fit would be also possible here and the slope would be completely different. Please provide a motivation for your approach because otherwise the calculated activation energy might be somewhat arbitrary.

Response: The approach is actually valid for the 1.5 microns sample of the Au based alloy too, where the points at the lowest cooling rates run parallel to those at the same cooling rates of the 0.9 microns sample. To some extent, this is also valid for the points at the lowest cooling rates for the 2 microns sample of the Pt based alloy, though data points are scarcer in this case. The largest samples, even at the lowest cooling rates, are in the crossover region between alpha to non-alpha controlled vitrification kinetics. Hence, the activation energy has not yet attained its low temperature steady state. To clarify this point, we have added the following text in the main manuscript:

“This is done only for the smallest samples, for which the crossover from α to non- α controlled regime is fully attained. If data at lower cooling rates were available, this could be done for larger samples too.”

REVIEWER COMMENTS

Reviewer #1 (Remarks to the Author):

The authors have addressed my comments, I recommend it for publication.

Reviewer #3 (Remarks to the Author):

The authors have addressed most the concerns and queries of the reviewers properly and clarified the open points. The manuscript has been revised gently according to these comments/queries.

Maybe the wording of some of my previous comments has not been sufficiently clear but I feel that the following points might have not been considered in a way they could have been. I do not want to appear pedantic but a slightly more elaborate discussion would have helped the manuscript to my mind. It would have put some of the values and statements into perspective. Though I do not intend to impede publication of the present manuscript, the authors might still want to consider incorporating these aspects in their work prior to publication.

My concerns are mainly related to the typical errors made in the determination of some of the data discussed in the manuscript:

1. My previous comment #3: Now you have included a short description in the main text referring to the Moynihan method for determining T_f . This is appreciated, however, SFig. 1 is already quite clear regarding the approach. The point, I originally made and I want to stress again is that the linear fit of c_p for the glass and the liquid should have a rather strong effect on the value of T_f . As SFig. 1 reveals, the glassy state can be fit rather well but the data for a linear fit of the liquid state is somewhat limited. A line with a different slope could be chosen as well, as it seems. Fig. 2a and b show that this might be an issue for all Au-based samples. Therefore, I had wished for a brief discussion/estimate of the error typically made in T_f . Most likely, the error will not be very dramatic and probably in the range of the symbol size in the according Arrhenius plot (Fig. 3 and b). But if you have the c_p -data of the melt for even higher temperatures it would be helpful to depict them (at least in SFig. 1). Your reply has not addressed this at all.

2. My previous comment #9: I agree that for the second smallest Au-based glass ($l_{eq} = 1.5 \mu\text{m}$) the number of data points is probably too limited for a reliable Arrhenius fit and therefore, it is disputable whether or not the Arrhenius fit would yield the same slope as for the smallest sample. In case of the Pt-based glass (Fig. 3b) the slopes of the smallest ($l_{eq} = 1.3 \mu\text{m}$) and the second smallest sample ($l_{eq} = 2.0 \mu\text{m}$) are most likely quite different, which means the activation energy of the fast vitrification processes are different. A slightly more critical discussion of this aspect would have been helpful because it helps to understand the significance/reliability of the calculated activation energy.

The authors have addressed most the concerns and queries of the reviewers properly and clarified the open points. The manuscript has been revised gently according to these comments/queries.

Response: We thank the Reviewer for recognizing the effort we made to respond to his/her comments.

Maybe the wording of some of my previous comments has not been sufficiently clear but I feel that the following points might have not been considered in a way they could have been. I do not want to appear pedantic but a slightly more elaborate discussion would have helped the manuscript to my mind. It would have put some of the values and statements into perspective. Though I do not intend to impede publication of the present manuscript, the authors might still want to consider incorporating these aspects in their work prior to publication.

Response: We understand the point made up by the Reviewer. Indeed, some points of our previous response may be unclear to many readers and, therefore, require further clarification.

My concerns are mainly related to the typical errors made in the determination of some of the data discussed in the manuscript:

1. My previous comment #3: Now you have included a short description in the main text referring to the Moynihan method for determining T_f . This is appreciated, however, SFig. 1 is already quite clear regarding the approach. The point, I originally made and I want to stress again is that the linear fit of c_p for the glass and the liquid should have a rather strong effect on the value of T_f . As SFig. 1 reveals, the glassy state can be fit rather well but the data for a linear fit of the liquid state is somewhat limited. A line with a different slope could be chosen as well, as it seems. Fig. 2a and b show that this might be an issue for all Au-based samples. Therefore, I had wished for a brief discussion/estimate of the error typically made in T_f . Most likely, the error will not be very dramatic and probably in the range of the symbol size in the according Arrhenius plot (Fig. 3 and b). But if you have the c_p -data of the melt for even higher temperatures it would be helpful to depict them (at least in SFig. 1). Your reply has not addressed this at all.

Response: The point we made in our response was not sufficiently clear and the previous version of Supplementary Fig. 1 was misleading. The liquid c_p was actually taken from linear response results, where the absence of non-reversing contributions to c_p and the possibility of varying the frequency over a wide range allowed us to obtain it in a temperature range of about 60 K. This is now clarified in the revised version of Supplementary Fig. 1, where, in the newly added right panel, we have incorporated $c_{p,rev}$ data (already present in the main manuscript) together with those of the left panel (obtained from standard heating ramps). The large range of experimentally available data of liquid c_p allows a reliable application of the Moynihan method and, therefore, determination of T_f . This result in a negligible error in the determination of T_f , smaller than 2 K. Apart from the new panel in Supplementary Fig. 1, we have added the following sentence in the main manuscript:

"...encompassing a wide range of temperature, which for the Δc_p is as large as 60 K."

And in the Supplementary Information:

“For a reliable determination of c_p^l and c_p^g , we used data from step response analysis, delivering the reversing specific heat. As shown in panel b of Supplementary Figure 1, above all for c_p^l , in doing so a wide temperature range, about 60 K, is considered for the linear fitting.”

2. My previous comment #9: I agree that for the second smallest Au-based glass ($l_{eq} = 1.5 \mu\text{m}$) the number of data points is probably too limited for a reliable Arrhenius fit and therefore, it is disputable whether or not the Arrhenius fit would yield the same slope as for the smallest sample. In case of the Pt-based glass (Fig. 3b) the slopes of the smallest ($l_{eq} = 1.3 \mu\text{m}$) and the second smallest sample ($l_{eq} = 2.0 \mu\text{m}$) are most likely quite different, which means the activation energy of the fast vitrification processes are different. A slightly more critical discussion of this aspect would have been helpful because it helps to understand the significance/reliability of the calculated activation energy.

Response: We agree with the Reviewer that the Arrhenius fit and the consequent obtainment of the activation energy can be reliably done only for the smallest samples. For the largest, the number of data points is too limited as, with increasing the temperature, the crossover to highly activated alpha relaxation regime begins. We have clarified this point modifying manuscript with the following text:

“For larger samples, the range of data where the Arrhenius fit can be reliably performed is either too limited or absent, due to the incipient crossover to the highly activated α relaxation regime.”

REVIEWERS' COMMENTS

Reviewer #3 (Remarks to the Author):

Thank you for your reply and answers. I do not have any additional comments or queries.